# Full-Length Transcriptome Maps of Reef-Building Coral Illuminate the Molecular Basis of Calcification, Symbiosis, and Circadian Genes

**DOI:** 10.3390/ijms231911135

**Published:** 2022-09-22

**Authors:** Tingyu Han, Xin Liao, Yunchi Zhu, Yunqing Liu, Na Lu, Yixin Li, Zhuojun Guo, J.-Y. Chen, Chunpeng He, Zuhong Lu

**Affiliations:** 1State Key Laboratory of Bioelectronics, School of Biological Science and Medical Engineering, Southeast University, Nanjing 210096, China; 2Guangxi Key Laboratory of Mangrove Conservation and Utilization, Guangxi Mangrove Research Center, Beihai 536000, China; 3Nanjing Institute of Paleontology and Geology, Nanjing 210008, China

**Keywords:** reef-building corals, Symbiodiniaceae, holobionts, PacBio Sequel II, full-length transcriptome, gene expression profile, biomineralization, symbiosis, circadian clock

## Abstract

Coral transcriptomic data largely rely on short-read sequencing, which severely limits the understanding of coral molecular mechanisms and leaves many important biological questions unresolved. Here, we sequence the full-length transcriptomes of four common and frequently dominant reef-building corals using the PacBio Sequel II platform. We obtain information on reported gene functions, structures, and expression profiles. Among them, a comparative analysis of biomineralization-related genes provides insights into the molecular basis of coral skeletal density. The gene expression profiles of the symbiont Symbiodiniaceae are also isolated and annotated from the holobiont sequence data. Finally, a phylogenetic analysis of key circadian clock genes among 40 evolutionarily representative species indicates that there are four key members in early metazoans, including *cry* genes; *Clock* or *Npas2*; *cyc* or *Arntl*; and *tim*, while *per*, as the fifth member, occurs in Bilateria. In summary, this work provides a foundation for further work on the manipulation of skeleton production or symbiosis to promote the survival of these important organisms.

## 1. Introduction

Coral reefs are among the most productive and biodiverse ecosystems on Earth [1], and they provide survival habitats for approximately 30% of marine life [2,3,4,5]. Approximately 500 million people worldwide depend on these reefs, demonstrating their great ecological and economic value [6,7,8,9,10]. The scleractinian corals that mainly produce coral reefs constitute an important branch of the metazoan [2]. They show a body plan typical of Phylum Cnidaria [9] and retain many ancient gene features of ancestral metazoans [11], providing an important genetic background for studying the evolutionary origin of metazoans and bilaterians [12,13]. At present, our understanding of the molecular biology of reef-building corals largely derives from omics sequencing, and these data have provided crucial information on coral calcification [14,15], symbiosis [16,17,18,19,20,21], heat stress [22,23,24], acid stress [25,26], cnidocytes [27,28], collagen secretion [29], immunity [28,30,31,32], budding [33,34], and circadian clocks [35,36,37]. Despite these advances, many open questions remain.

The greatest contribution of scleractinian corals to marine biota and ecosystems is the deposition of reefs via biomineralization in calicoblasts [28], which form skeletons. The reef-building coral skeleton is built through the continuous deposition of aragonite [38], which is formed by a mineral fraction consisting of calcium carbonate and an organic matrix molecule fraction that includes carbohydrates, lipids, and proteins [14]. Current reports indicate that the main components used for calcification include calcium ATPase, carbonic anhydrase (CA), bicarbonate transporter (i.e., solute carriers 4 [SLC4] and 26 [SLC26]) and core skeleton organic matrix proteins (SOMPs) (e.g., acid-rich protein, uncharacterized skeletal organic matrix proteins [USOMPs], galaxins, and alpha IV collagen) [14,15,38,39,40,41,42,43,44,45,46,47,48,49]. However, due to the lack of full-length sequences for these genes, in-depth biological studies cannot be performed.

Phagocytosis targeting primitive algae was a prerequisite for the development of food webs involving multicellular animals [50,51,52,53]. Additionally, paleontological [51], geochemical [50,51], and molecular clock [50,51,52] evidence suggests that various forms of eukaryotic predation occurred during the Neoproterozoic Era (1000–541 million years ago), from the proliferation of marine algae to the origin of multicellular animal clades. At that time, increasing algal abundance created food webs with more efficient nutrient and energy transfers, driving shifts in ecosystems towards larger and increasingly complex organisms [50]. This effect is recorded in the appearance of sponges and the subsequent radiation of eumetazoans in the Ediacaran period [50]. At present, most hosts of intracellular symbionts belong to groups with polyp body plans, such as hydra, soft corals, anemones, and reef-building corals, and Mollusca, such as giant clams [16,17,18,19,20,21,27,54,55,56]. Among these groups, reef-building corals have particularly widespread and extensive endosymbiosis (with Symbiodiniaceae). Usually, intracellular symbiosis is thought to depend on molecular recognition for the acquisition of algae [57]. Symbiodiniaceae, similar to other dinoflagellates, must produce low doses of dinotoxins as a defense against predation or as metabolic byproducts; hence, host cells must perform certain reactions to reduce algal toxicity in an intracellular symbiotic system [58,59,60]. An incompatible dynamic equilibrium relationship with intracellular algae may lead to host cell death, and reducing immune rejection is a method for alleviating immune conflict and benefiting from these algae during intracellular symbioses [54]. Therefore, it is worth further exploring what the corals are obtaining by living with these algae, and whether there are differences in gene expression levels of what they supplied to different corals.

Reef-building corals show obvious circadian rhythms; interestingly, the circadian clock system and related genes have not been thoroughly investigated, although they play important roles in sustaining healthy coral growth [35,36,37]. Although there is a debate, which has been especially intense since 2008, on whether sponges or ctenophores are the earliest branching animals [61,62,63,64,65,66,67,68], both of them have identified some core circadian clock genes, reflecting the existence of circadian clock system in the last common animal ancestor [69,70,71]. At the basic molecular level, the operations of the circadian clock system and biological rhythm behavior are regulated through a conserved negative transcription-translation feedback loop, which is controlled by five key gene families [72]. Cryptochromes (CRYs) are a class of flavoproteins that can detect blue light [73]. Generally, circadian locomotor output cycles kaput (CLOCK) and its homologous protein neuronal PAS domain-containing protein 2 (NPAS2), as well as cycle (CYC) and its homologous protein brain and muscle ARNT-like (BMAL), act as positive regulatory factors, while period (PER) and timeless (TIM) act as negative regulatory factors [74]. However, the specific connections among these families vary across clades. In *Drosophila*, CRY regulates the circadian clock in a light-dependent manner [75], whereas in mice, CRY1 and CRY2 act as light-independent inhibitors of the CYC-CLOCK component [76]. However, in some invertebrates, such as monarch butterflies, CRYs exhibit both *Drosophila*-like and mammal-like functions, providing evidence of an ancestral clock gene regulation state [77,78]. Three *cry* genes and fifteen other homologous bilaterian circadian clock genes have been found in the *Acropora digitifera* coral genome [35], and 24 genes have been matched to 6 insect and 18 mammal key circadian genes via NCBI BLAST in the *Acropora millepora* coral transcriptome [36,37]. The composition and related phylogenetic pattern of circadian genes among the Anthozoa lineage remain unclear; more omics data are needed to elucidate them.

On a global scale, *Pocillopora damicornis*, *Acropora muricata*, and *Montipora foliosa* are common dominant reef-building corals in the Indo-Pacific region, and *Pocillopora verrucosa* is a typical reef-front stony coral that protects fringing reefs [79,80,81,82]. These four corals play pivotal roles in the Indo-Pacific coral reef ecological system. Recently, increasing amounts of sequencing data derived from these reef-building corals have been deposited in public databases (see Table 1 and Appendix A). However, all currently available sequences have been generated using short-read (50–300 bp) sequencing approaches, leaving incomplete information and sequence splicing errors [83,84,85]. Critically, although some high-quality sequences have been published for both corals themselves [23,24,30,31,32,86,87,88,89,90,91,92,93,94,95,96,97,98,99,100,101] and their associated microorganisms [33,102,103,104,105,106,107,108,109,110,111,112,113,114,115,116,117,118,119,120,121], the reliance on short reads has prevented the precise delimitation of the gene expression profiles of reef-building corals and their endosymbiotic Symbiodiniaceae. Recent advances in long-read sequencing technology (e.g., PacBio Sequel II) have made it possible to obtain large amounts of full-length transcript data from many organisms and tissues [122,123,124]. In principle, such data should permit the characterization of all expressed transcripts as complete, contiguous mRNA sequences from the transcription start site to the transcription end site; in turn, this enables the more accurate and efficient analysis of the full spectrum of gene expression profile information, including data on gene expression, alternative splicing, gene fusion, expression regulation, coding sequences (CDSs), and protein structure [125,126,127,128].

To address these questions, we apply a full-length transcriptomic isoform sequencing (Iso-Seq) strategy using the PacBio Sequel II sequencing platform and quantitative gene expression analysis using the Illumina HiSeq X Ten sequencing platform to produce transcriptome maps for the four aforementioned important reef-building corals and their endosymbiotic Symbiodiniaceae. Based on the gene expression profiles, we perform a more thorough analysis of several important physiological features of reef-building corals, including biomineralization-related gene groups, circadian gene families, and coral and Symbiodiniaceae factors related to the endosymbiotic interaction. These results enhance our understanding of the molecular biology and ecology of reef-building corals, which may aid in the recovery of marine ecosystems, and provide insights into the evolution of the circadian circuitry and holobiosis.

## 2. Results

### 2.1. Full-Length-Enriched Transcriptome Sequencing and Data Processing

Based on the standard processes of the PacBio Sequel II sequencing platform, the full-length-enriched raw transcriptome sequencing data (polymerase reads) of four coral holobionts obtained included 25.34 Gb in *P. damicornis*, 27.8 Gb in *P. verrucosa*, 22.32 Gb in *A. muricata*, and 21.44 Gb in *M. foliosa* (Table 2 and Appendix A). These data were filtered to ensure quality and reliability, including removing reads of less than 50 bp in length and adapter sequences (for details and statistics, see Table 2 and Appendix A, Appendix A). The gene expression profiles of corals and their symbiotic Symbiodiniaceae (whose transcripts were analyzed separately in a later subsection) were separated based on an alignment to previously published sequences. In terms of the corals themselves, there were 20,609 transcripts and 14,167 unigenes (N50 = 2954 bp) in *P. damicornis*, 24,174 transcripts and 12,822 unigenes (N50 = 2313 bp) in *P. verrucosa*, 31,242 transcripts and 13,800 unigenes (N50 = 2126 bp) in *A. muricata*, and 25,460 transcripts and 10,905 unigenes (N50 = 1678 bp) in *M. foliosa* (Table 2 and Appendix A). Unigene lengths were concentrated in the range of 1–3 kbp, with few unigenes of less than 1 kbp, verifying that the low-quality sequencing data had been filtered out (Appendix A and Appendix A). A previous study reported that there were 297,221 assembled transcripts (N50 = 1831 bp) and 209,337 unigenes (N50 = 1435 bp) in *P. damicornis* using a short-read sequencing approach [88]. It is obvious that PacBio sequencing can obtain fewer redundant sequences and higher-quality sequences, although we sequenced only one developmental stage of coral and not all transcripts were obtained. Then, to evaluate the level of redundancy in the data, we examined the unigene-to-transcript ratio. The percentages of unigenes with a one-to-one unigene-to-transcript ratio were 73.71% (*P. damicornis*), 64.75% (*P. verrucosa*), 57.15% (*A. muricata*), and 57.26% (*M. foliosa*), and the percentages of unigenes with a one-to-two unigene-to-transcript ratio were 17.44% (*P. damicornis*), 17.52% (*P. verrucosa*), 18.41% (*A. muricata*), and 17.18% (*M. foliosa*) (detailed in Appendix A and Appendix A). The sum of the two ratios was more than 74% in all samples, indicating that data redundancy was largely reduced. Thus, we obtained high-quality full-length-enriched transcriptome sequencing data for four coral holobionts that were suitable for the subsequent analysis.

### 2.2. Gene-Function Annotation and Structure Analysis

To obtain comprehensive gene-function information, seven authoritative databases (NR [129], NT, Pfam [130], KOG [131], Swiss-Prot [132], KEGG [133], and GO [134]) were utilized to annotate the unigenes, and the related statistics are shown in Figure 1 (Appendix A and Appendix A). In the four corals, 93.88% (*P. damicornis*), 89.32% (*P. verrucosa*), 95.10% (*A. muricata*), and 79.64% (*M. foliosa*) of the unigenes were annotated in at least one database (Figure 1a and Appendix A). The percentage of annotated unigenes in *A. muricata*, *P. damicornis*, and *P. verrucosa* was approximately 90%, suggesting that most of the unigenes in these species were orthologues of genes with functional annotations available. In contrast, the percentage of unigenes annotated in *M. foliosa* was only 79.64% because sequencing data obtained from *Montipora* are rare, and the adaptive specialized evolution of these species means that their genomes show especially great differences from those of other reef-building corals with available omics data [100,135], limiting their in-depth annotation. It was also obvious that the percentage of annotated unigenes in *A. muricata* was higher than that in the other three corals (Figure 1b) because *A. digitifera* [136] and *A. millepora* [137], with reported genome data, both belong to the same genus as *A. muricata*, providing a good reference genome background.

To explore whether the sequences we obtained were indeed derived from cnidarians rather than from some contaminants, we counted the number of sequences annotated based on species information and observed that the top five species with the most highly similar genes to those of the four investigated corals were *A. digitifera*, *Exaiptasia pallida*, *Nematostella vectensis*, *Branchiostoma belcheri*, and *Stylophora pistillata* (or *A. millepora*) (Figure 2, Table 3 and Appendix A). These species (other than the amphioxus *B. belcheri*) are all cnidarians, and the similarities of the investigated corals to the reference species were all greater than 93%, indicating the relatively high accuracy of the annotations. Interestingly, the NR results indicate that the gene expression profiles of all four investigated corals are close to that of *B. belcheri* (Figure 2, Table 3 and Appendix A), which may be due to the slow evolution of amphioxus protein sequences.

The prediction of CDSs is helpful for potential unigene analysis and provides a basis for subsequent protein analysis. The CDS lengths of the four corals were concentrated in the range of 300–1700 nt (Materials and Methods Section and Appendix A), and totals of 97.91% (*P. damicornis*), 95.67% (*P. verrucosa*), 95.51% (*A. muricata*), and 89.04% (*M. foliosa*) of the unigenes were predicted to contain CDS regions (Appendix A). We then identified coral TFs among the predicted protein CDS results based on Pfam TF families (Materials and Methods Section) and found that the five largest TF families in the corals were ZBTB, zf-C2H2, homeobox, bHLH, and TF_bZIP (Appendix A and Appendix A). The results of the simple sequence repeat (SSR) detection showed that most SSR sequences corresponded to poly(A) tails (Appendix A and Appendix A), reconfirming the absence of genomic DNA contamination. Transcripts without coding potential were classified as lncRNAs. Generally, lncRNAs were shorter than mRNAs (Appendix A). The number of predicted lncRNAs in *M. foliosa* was much higher than those in the other corals (Appendix A). Following CDS, TF, SSR, and lncRNA analyses, we quantified the gene expression profiles in each coral by mapping the Illumina sequencing reads to the full-length transcriptome data. This mapping directly providing read-count values that could be converted to the expected number of fragments per kilobase of transcript sequence per million base pairs sequenced (FPKM) or transcripts per kilobase million (TPM) values for further ensuring diverse quantitative analysis under different conditions (Appendix A). Based on these data, the Pearson’s correlation coefficient (R) analysis can be used to reflect the gene expression profile similarity among the samples, and thus verify the experimental reliability and rationality of the sample selection.

### 2.3. Gene Expression Profile Analysis

We then analyzed the gene expression profiles of the four corals (Figure 3a–d). It was clear that the gene expression levels of each coral were considered to be consistent across all replicates (Figure 3a and Appendix A). In addition, a Pearson’s R^2^ > 0.92 among the three biological replicates of each coral suggested that the sample selection was reasonable and the experimental data were reliable (Figure 3d). Then, the differences in the coral gene expression profiles were explored. We used two methods (median [138] and scbn [139]) to analyze the pairwise differential expressions between the two corals and showed that all six groups produced significant differential expressions of transcript orthologs (|log2FoldChange| > 2 or 10; *p*-value < 10^−6^; see Materials and Methods Section for complete statistical analysis, Appendix A and Appendix A). Among them, the highest number of orthologous transcripts was observed between *P. damicornis* and *P. verrucosa*, while their differentially expressed genes (DEGs) accounted for the smallest proportion of all orthologous transcripts (Appendix A and Appendix A). Subsequently, according to the gene expression profile results, biomineralization, symbiosis, and circadian gene families, which provide crucial information on coral population structure and the evolution of coral gene repertoires, are highlighted hereafter.

### 2.4. Expression Analysis of the Biomineralization-Related Gene Group

Although the biomineralization mechanisms of reef-building corals are similar (Figure 4a), they are displayed with a wide range of morphological differences. To obtain an insight into the molecular-level causes of this phenomenon, we investigated the expression patterns and features of biomineralization-related gene families and observed that the expression levels of calcium ATPases, which are essential for Ca^2+^ transport, were higher in the genera *Pocillopora* and *Acropora*, and the expression levels of bicarbonate transporters, which are essential for HCO_3_^−^ transport, were higher in *P. verrucosa* (Figure 4b,c, Table 4 and Appendix A). Among the four investigated corals, *M. foliosa* had the lowest expression level of calcium ATPases, which may have a negative effect on the transportation of calcium ions to the calcifying fluid (Figure 4b, Table 4 and Appendix A). CA2 is universally expressed in all four corals at very high levels and accounts for a major proportion of the CA content, implying its key role in coral biomineralization (Figure 4d, Table 4 and Appendix A).

The expression levels of coral biomineralization-related extracellular matrix (ECM) proteins, including acid-rich proteins, USOMPs, galaxins, and collagen alpha-6 (VI) chain proteins, were all obviously higher in the Complexa clade (*A. muricata* and *M. foliosa*) than in the Robusta clade (*P. damicornis* and *P. verrucosa*) (Figure 4e–g, Table 4 and Appendix A). Regarding key acid-rich ECM proteins, *A. muricata* showed the highest expression levels, diversity, and balance, followed by *M. foliosa*, while skeletal aspartic acid-rich protein 1 (SAARP1) was the only protein in this group highly expressed in genus *Pocillopora*. Regarding the ECM accessory proteins USOMPs, *A. muricata* and *M. foliosa* presented extremely higher levels and diversity, and USOMP-6 was the dominant *USOMP* gene, whereas members of the genus *Pocillopora* mainly expressed USOMP-5 (Table 4 and Appendix A). Galaxins were the only ECM components with a higher expression in *M. foliosa* than in *A. muricata* (Table 4 and Appendix A). This result indicates that there are significant differences in the types and expression levels of core calcification genes among different reef-building corals, which may affect their morphological formation and growth model.

### 2.5. Genes Expressed by Symbiodiniaceae (Intracellular Symbionts of Coral) Are Mainly Involved in Energy and Nutrient Production

The secretion of calcium carbonate skeletons to form marine reefs and the reliance on intracellular symbiotic algae (Symbiodiniaceae) for nutrient acquisition are two signature features of reef-building coral physiology [14,15,16,17,18,19,20,21,136]. Although skeletal formation has been extensively discussed, the contributions of the genes expressed by the Symbiodiniaceae are rarely mentioned. Depending on identifying coral or symbiont transcripts based on alignment to previously published sequences, we found that 235 (*P. damicornis*), 351 (*P. verrucosa*), 220 (*A. muricata*), and 561 (*M. foliosa*) Symbiodiniaceae unigenes were expressed in the investigated coral holobionts (Appendix A). Similar to the analysis process used for corals, Symbiodiniaceae gene expression profiling was performed using seven functional databases, NR, NT, Pfam, KOG, Swiss-Prot, KEGG, and GO, and published Symbiodiniaceae genome data (Appendix A and Appendix A.). We found 832 transcripts that were expressed by the symbiotic Symbiodiniaceae of all four investigated corals (Appendix A), accounting for approximately 70.75% (*P. damicornis*), 70.87% (*P. verrucosa*), 88.79% (*A. muricata*), and 93.69% (*M. foliosa*) of the symbiont transcripts. Overall, gene expression profiles of the Symbiodiniaceae were more similar than those of coral hosts, suggesting considerable gene expression convergence. On the other hand, the gene expression patterns of the symbionts of *P. damicornis* and *P. verrucosa* were more similar to each other than to *A. muricata* and *M. foliosa*, and vice versa, coinciding with the evolutionary relationships of the corals (Appendix A).

To explore the functions of these expressed genes, we classified them according to the annotation results using the GO database (Figure 5 and Appendix A, and Appendix A). In the biological process category, the identified gene functions were mainly involved in oxidation reduction and carbohydrate metabolism (Figure 5). In the cellular component category, most of the functional genes were related to photosynthesis, such as light-harvesting complex or photosystem (Appendix A). In the molecular function category, the largest group of genes was associated with oxidoreductase or carbonate dehydratase activity (Appendix A). These results suggest that the symbionts of all four corals have similar expression profiles that feature genes related to energy and nutrient production.

### 2.6. Phylogenetic Analysis of the Key Circadian Clock Gene Regulation Network

The circadian clock system is one of the most universal and fundamental characteristics of life across almost all living species [74]. Corals also exhibit circadian behaviors, and there have been studies describing the molecular mechanisms underlying the regulation of these behaviors [35,36,37]. However, whether this mechanism is applicable to other corals, other cnidarians and even other animals remains to be clarified. To explore the evolutionary origins of the key members of the circadian clock gene regulation network of reef-building corals, including the four corals studied herein, we selected 40 evolutionarily representative species and used their protein models to investigate the key circadian clock gene regulation network, namely, the *cry1*, *cry2*, *Clock, Npas2*, *cyc, Arntl* (also known as *Bmal1*), *Arntl2* (also known as *Bmal2*), *per1*, *per2*, *per3*, and *tim* genes (Table 5 and Appendix A), and performed phylogenetic analyses (Figure 6 and Appendix A). The results show that the key circadian clock gene regulatory network evolves in early metazoans, whereas protozoan clade does not have this kind of gene regulatory network, although the *tim* orthologous gene is found in the Ciliophora clade. Moreover, the Ascidiacea clade showed a complete loss of this network during evolutionary adaptation, except for the *tim* gene, suggesting that ascidians might have unknown clock mechanisms that differ from known systems of vertebrates and insects (Table 5 and Appendix A, Figure 6 and Appendix A). Among this group of genes, the *tim* gene was the only one present in all the searched species, suggesting that it is the most conserved and stable circadian clock gene (Table 5 and Appendix A, Figure 6) [100,140]. The phylogenetic analysis suggested that in the early metazoan stage (non-bilaterian metazoans), there were four key gene members of the circadian clock gene regulatory network, *cry*; *Clock* or *Npas2*; *cyc*, *Arntl*(*Bmal1*), or *Arntl2*(*Bmal2*); and *tim*. The *cry*, *Clock* or *Npas2*, cyc or *Arntl* (*Bmal1*), and *tim* gene homologues can be found in reef-building corals as shown in Table 5. Thus, the data suggested the existence of four key conserved members of the reef-building coral circadian clock gene regulation network. The phylogenetic analysis also indicated that the *per* gene is a novel member of the circadian clock gene regulation network in the Bilateria, as it is found in protostomes and deuterostomes but not in early metazoans (Table 5).

## 3. Discussion

Despite much research, our understanding of reef-building coral holobiont transcriptomes remains incomplete, and the short-read lengths intrinsic to the prevailing technologies have limited access to complete genetic information. Here, we sequenced and analyzed the full-length transcriptomes of four common dominant reef-building coral holobionts with respect to gene functions, structures, and expression. The results reveal differences in the members actually involved in biomineralization processes in reef-building corals and provide new insights into further understanding the molecular mechanisms of coral density. In particular, we isolated and demarcated the gene expression profiles of the symbiont Symbiodiniaceae, which showed a higher convergence than the coral hosts, suggesting that the internal environment of symbiotic algae-bearing cells in all four species may be quite similar. Moreover, we confirmed that there were four key members of the circadian clock gene regulatory network in early metazoans and that the *per* gene first occurred in Bilateria, further enriching the molecular database for studying the evolutionary origins of the animal circadian clock system, although this needs to be verified by biological experiments or other methods.

### 3.1. Coral Biomineralization and Skeleton Density

Coral biomineralization has been intensively studied in previous studies, but descriptions of its formation mechanism vary; here, we summarized this process, which consists of four components (Figure 4a). (1) Calcium ion transport. In calicoblasts and paracells, calcium ions enter the cells through calcium channels and exit via calcium ATPases that exchange two calcium ions for four protons across the cell membrane [39,40]. Ca^2+^ diffusion among cells with chemical gradients may also participate in calcium deposition [15]. (2) An HCO_3_^−^ source. The source of HCO_3_^−^ is metabolic and environmental CO_2_. Metabolic CO_2_ can be converted into HCO_3_^−^ both intracellularly and extracellularly, which is catalyzed by CA under a favorable pH [41,42,43], and intracellular HCO_3_^−^ exits cells via bicarbonate transporters belonging to two membrane protein families (SLC4 and SLC26) [44]. In calcifying fluid, the HCO_3_^−^ concentration is higher than the Ca^2+^ concentration, and the amount of calcium carbonate deposition is determined by the Ca^2+^ concentration [43]. (3) Acid-rich proteins. Coral acid-rich proteins are key ECM proteins involved in biomineralization and can interact with amorphous calcium carbonate (ACC) directly, promoting crystal nucleation, determining growth axes and controlling crystal growth [45,46]. (4) Other organic matrix proteins of the ECM. The bioprecipitation of aragonite crystals in corals also requires additional skeletal organic matrix proteins as binders [47], including USOMPs, galaxins, and alpha IV collagen [38,48,49]. In addition, magnesium plays a crucial role in regulating the formation of different calcium carbonate phases that cooperate with organic matrix molecules to stabilize ACCs [141].

Here, the above core biomineralization-related gene expression level analyses among four reef-building corals indicate that the genera *Pocillopora* and *Acropora* may have a greater capacity to produce calcium carbonate than the genus *Montipora*, but the opposite situation was observed for biomineralization-related ECM proteins from the genus *Pocillopora*, suggesting that the species of the Complexa clade may produce a greater volume of skeleton than those of the Robusta clade using the same amount of calcium carbonate. This is in accordance with the reported skeletal density data of these stony corals [34,142]; however, future studies on the relationship between the activity and stability of biomineralization-related enzymes among different corals at the protein level are required to confirm this conjecture.

### 3.2. Evolutionary Origins of the per Gene

The oscillations of both the transcript and protein levels of the *per* gene have a period of approximately 24 h and play a central role in the molecular mechanism of the biological clock driving circadian rhythms. In *Drosophila*, after PER is produced from *per* mRNA, it dimerizes with timeless (TIM), and the complex enters the nucleus and inhibits the TFs *per* and *tim*, which in turn lowers the levels of PER and TIM [53]. When TIM is not complexed with PER, another protein, double-time, or DBT, phosphorylates PER, targeting it for degradation [57]. In mammals, an analogous transcription-translation negative feedback loop is observed. One of the three PER proteins (PER1, PER2, and PER3) that is translated from the mammalian homologs of drosophila-per dimerizes via its PAS domain with one of two cryptochrome proteins (CRY1 and CRY2) to form a negative element of the clock. This dimer then interacts with the CLOCK (or NPAS2) and ARNTL (or ARNTL2) heterodimer, inhibiting its activity and thereby negatively regulating its own expression [143]. Despite an extensive search, we failed to detect orthologs of the *per* gene in our coral data. This also may be the case for sponges and ctenophores (Table 5 and Appendix A). On the other hand, amphioxus [144,145] and *Crassostrea gigas* [146] likely have *per* genes, and our results show that the biological clock is a common feature of a wide range of Metazoa, while *per* gene originates from Bilateria. It is unclear, however, whether the ancient circadian clock system initially operated without the *per* gene, whether the non-bilaterian metazoans system lost this locus via deletion or other mutation, or whether unknown proteins may have functions similar to the *per* gene. These hypotheses need to be explored further in future research.

Overall, the full-length transcriptome maps of reef-building coral may serve as references for expression analyses of coral under environmental stressors linked to global change, which alter the normal function of reef-building corals. As long-read sequencing continues to evolve in throughput, accuracy, accessibility, and cost efficiency, full-length transcriptomes will be adopted by researchers and provide us with unprecedented views of serious biological problems. However, genome, small RNA, proteome, and single-cell data are still lacking to further understand the biology of reef-building corals. Future efforts are required to construct a coral genome database as a reference for analyzing functional genes and their associated regulatory networks at the transcriptomic and proteomic levels and further refine them to specific cellular lineages.

## 4. Materials and Methods

### 4.1. Ethics

All coral samples were collected and processed in accordance with local laws for invertebrate protection.

### 4.2. Sample Collection

The corals in the study were collected from the Xisha Islands in the South China Sea (latitude 15°40′–17°10′ N, longitude 111°–113° E).

### 4.3. Coral Culture System

The coral samples were cultured in our laboratory coral tank with conditions conforming to their habitat environment. All samples were raised in a RedSea^®^ tank (redsea575, Red Sea Aquatics Ltd., London, UK) at 26 °C and 1.025 salinity (Red Sea Aquatics Ltd., London, UK). The physical conditions of the coral culture system were as follows: three coral lamps (AI^®^, Red Sea Aquatics Ltd., London, UK), a protein skimmer (Reef Octopus Regal 250-S, Honya Co. Ltd., Shenzhen, China), a water chiller (tk1000, TECO Ltd., Port Louis, Mauritius), two wave devices (VorTech™ MP40, EcoTech Marine Ltd., Bethlehem, PA, USA), and a calcium reactor (Reef Octopus OCTO CalReact 200, Honya Co. Ltd., Shenzhen, China).

### 4.4. Total RNA Extraction

The three biological replicates samples for each coral were isolated from three healthy branches in the same coral independent colony to ensure that enough high-quality RNA (>15 µg) could be obtained for a PacBio cDNA library and three Illumina cDNA libraries. All the RNA extraction procedures followed the manufacturer’s instructions. The total RNA was isolated with TRIzol^®^ LS Reagent (Thermo Fisher Scientific, 10296028, Waltham, MA, USA) and treated with DNase I (Thermo Fisher Scientific, 18068015, Waltham, MA, USA). The high-quality mRNA was isolated with a FastTrack MAG Maxi mRNA Isolation Kit (Thermo Fisher Scientific, K1580-02, Waltham, MA, USA). The RNA extraction procedure was performed according to the following instructions: (1) grinded coral samples (kept the samples submerged in liquid nitrogen at all times); (2) when the samples were ground into small pieces, the TRIzol^®^ LS reagent was added, the ratio of sample to reagent was about 1:3; (3) we let samples stand and thaw naturally; (4) we continued adding TRIzol^®^ LS reagent until the samples were dissolved, and dispensed into 50 mL centrifuge tubes; (5) centrifuged at 4 °C and 3000 rpm for 5–15 min; (6) dispensed the supernatant into 50 mL centrifuge tubes; (7) added BCP (Molecular Research Center, BP 151, Cincinnati, OH, USA) to the above centrifuge tubes, the ratio of sample to reagent was about 5:1, shaken well and then stood for 10 min; (8) centrifuged at 4 °C and 10,500 rpm for 15 min; (9) we obtained the supernatant, added an equal volume of Isopropanol (Amresco, 0918-500ML, Radnor, PA, USA) and mixed well, stood them overnight at −20 °C; (10) centrifuged at 4 °C and 10,500 rpm for 30 min, discarded the supernatant; and (11) rinsed them 2 times with 75% Ice Ethyl alcohol, Pure (Sigma-Aldrich, E7023-500ML, Taufkirchen, München, Germany). Finally, three samples of each coral were extracted in equal amounts (total >10 µg) and mixed for PacBio full-length-enriched transcriptome sequencing; the remainders (>1.5 µg per sample) were used for Illumina sequencing.

### 4.5. Total RNA Quality Testing

Before establishing the library, the quality of total RNA must be tested. RNA degradation and contamination were monitored on 1% agarose gels electrophoresis; RNA purity (OD260/280 ratio) was checked using the NanoPhotometer^®^ spectrophotometer (IMPLEN, Westlake Village, CA, USA); RNA concentration was quantified using Qubit^®^ RNA Assay Kit in Qubit^®^ 2.0 Flurometer (Thermo Fisher Scientific, Waltham, MA, USA); and RNA integrity was assessed using the RNA Nano 6000 Assay Kit of the Agilent Bioanalyzer 2100 system (Agilent Technologies, Santa Clara, CA, USA).

### 4.6. Illumina cDNA Library Construction and Sequencing

A total amount of 1.5 µg RNA per sample was used as input material for the RNA sample preparations. Sequencing libraries were generated using NEBNext^®^ Ultra™ RNA Library Prep Kit (E7530L) for Illumina^®^ (NEB, Ipswich, MA, USA) following the manufacturer’s recommendations, and index codes were added to attribute sequences to each sample. Briefly, mRNA was purified from total RNA using poly-T oligo-attached magnetic beads. Fragmentation was performed using divalent cations under an elevated temperature in NEBNext First-Strand Synthesis Reaction Buffer (5×). First-strand cDNA was synthesized using a random hexamer primer and M-MuLV Reverse Transcriptase (RNase H^−^). Second-strand cDNA synthesis was subsequently performed using DNA Polymerase I and RNase H. Remaining overhangs were converted into blunt ends via exonuclease/polymerase activities. After adenylation of 3′ ends of DNA fragments, an NEBNext Adaptor with hairpin loop structure was ligated to prepare for hybridization. In order to select cDNA fragments preferentially 250–300 bp in length, the library fragments were purified with an AMPure XP system (Beckman Coulter, Beverly, Brea, CA, USA). Then, 3 µL USER Enzyme (NEB, Ipswich, MA, USA) was used with size-selected, adaptor-ligated cDNA at 37 °C for 15 min followed by 5 min at 95 °C before PCR. Then, PCR was performed with Phusion High-Fidelity DNA polymerase, Universal PCR primers, and Index (X) Primer. Finally, PCR products were purified (AMPure XP system) and library quality was assessed on the Agilent Bioanalyzer 2100 system. The clustering of the index-coded samples was performed on a cBot Cluster Generation System using TruSeq PE Cluster Kit v3-cBot-HS (Illumia, San Diego, CA, USA), according to the manufacturer’s instructions. After cluster generation, the library preparations were sequenced on an Illumina HiSeq X Ten platform and paired-end reads were generated.

### 4.7. PacBio cDNA Library Construction and Sequencing

The isoform sequencing (Iso-Seq) library was prepared according to the isoform sequencing protocol (Iso-Seq) using the Clontech SMARTer^®^ PCR cDNA Synthesis Kit (Clontech Laboratories (now Takara Laboratories), 634926, Mountain View, CA, USA) and the BluePippin Size Selection System protocol, as described by Pacific Biosciences (PN 100-092-800-03). Briefly, Oligo(dT)-enriched mRNA was reversely transcribed to cDNA by a SMARTer PCR cDNA Synthesis Kit; the synthesized cDNA was then amplified by polymerase chain reaction (PCR) using BluePippin Size-Selection System protocol; the Iso-Seq library was constructed by full-length cDNA damage repair, terminal repair, and attaching SMRT dumbbell adapters; the sequences of the unattached adapters at both ends of the cDNA were removed by exonuclease digestion; the cDNA obtained above was combined with primers and DNA polymerase to form a complete SMRT bell library. While the library was qualified, the PacBio Sequel II platform was used for sequencing based on the effective concentration and data output requirements of the library.

### 4.8. Data Filtering and Processing

The Illumina sequencing raw reads of fastq format were firstly processed through in-house perl scripts. In this step, clean data were obtained by removing reads containing adapters, reads containing ploy-N, and low-quality reads from raw data. At the same time, Q20, Q30, GC-content, and sequence duplication levels of the clean data were calculated. All the downstream analyses were based on clean data with a high quality.

The PacBio sequencing raw data were processed by SMRTlink v8.0 (Pacbio, Menlo Park, CA, USA) software. The circular consensus sequence (CCS) was generated from subread BAM files with the following parameters: min_length 50, min_passes 1, max_length 15,000. CCS.BAM files were output, which were then classified into full-length and non-full-length reads using lima, removing polyA using refine. Full-length fasta files produced were then fed into the cluster step, which performed isoform-level hierarchical clustering (n × log(n)), followed by final arrow polishing, hq_quiver_min_accuracy 0.99, bin_by_primer false, bin_size_kb 1, qv_trim_5p 100, and qv_trim_3p 30.

### 4.9. Coral and Symbiodiniaceae Sequences Separation

We aligned consensus reads to coral or Symbiodiniaceae reference genomes, respectively, using GMAP v2017-06-20 (Thomas D. Wu, South San Francisco, CA, USA) software [147]. The sequences mapped to Symbiodiniaceae reference genomes belonged to Symbiodiniaceae sequences; the sequences mapped to coral reference genomes belonged to coral sequences.

### 4.10. Correction and De-Redundancy

The RNA-seq data sequenced by the Illumina HiSeq X Ten platform were used to correct additional nucleotide errors in polish consensus sequences obtained in the previous step with LoRDEC v0.7 (Leena Salmela and Eric Rivals, Helsinki, Finland) software [148]. Using CD-HIT v4.6.8 (Weizhong Li, La Jolla, CA, USA) software (parameters: −c 0.95 −T 6 −G 0 −aL 0.00 −aS 0.99), all redundancies were removed in corrected consensus reads to acquire final full-length transcripts and unigenes for subsequent bioinformatics analysis [149].

### 4.11. Gene-Function Annotation

Gene functions were annotated using the following databases: NT (NCBI non-redundant nucleotide sequences); NR (NCBI non-redundant protein sequences); Pfam (protein family); KOG/COG database (clusters of orthologous groups of proteins); Swiss-Prot (a manually annotated and reviewed protein sequence database); KEGG (Kyoto Encyclopedia of Genes and Genomes); and GO (gene ontology). We used BLAST 2.7.1+ (Christiam Camacho, Bethesda, MD, USA) software [150] with the e-value ‘1×10^−5′^ for NT database analysis, Diamond v0.8.36 (Benjamin Buchfink, Tübingen, Germany) BLASTX software [151] with the e-value ‘1×10^−5′^ for NR, KOG, Swiss-Prot and KEGG databases analyses, and HMMER 3.1 (Sean R. Eddy, Ashburn, VA, USA) package [152] for Pfam database analysis.

### 4.12. Gene Structure Analysis

ANGEL v2.4 (Kana Shimizu, Tokyo, Japan) software [153] was used to predict protein CDS (coding sequence). We used same species or closely related species confident protein sequences for ANGEL training and then ran the ANGEL prediction for the given sequences. Usually, the TFs were identified based on the Pfam files of TF families in the AnimalTFDB 3.0 database [154]; however, corals were not included in this database, so we identified coral TFs based on the Pfam files of TF families using the hmmsearch program in HMMER 3.1 package. SSR of the transcriptomes was identified using MISA v1 (Thomas Thiel, Gatersleben, Germany) [155]. We used CNCI v2 (Liang Sun, Beijing, China) [156], CPC2 v0.1 (Yu-Jian Kang, Beijing, China) [157], PfamScan v1.6 (EMBL-EBI, Cambridgeshire, UK) [158], and PLEK v1.2 (Aimin Li and Junying Zhang, Xi’an, China) [159], four tools to predict the coding potential of transcripts. Transcripts predicted with coding potential by either/all of the three tools above were filtered out, and those lacking coding potential were our candidate set of lncRNAs.

### 4.13. Gene Expression Quantification

The full-length transcriptomes of each coral obtained above were used as the reference backgrounds, respectively, and then the clean reads for the corresponding three biological replicate samples obtained by Illumina sequencing were mapped to it and the read-count values for each transcript can be obtained by kallisto v0.44.0 (Pachter Lab, Berkeley, CA, USA) software [160]. The read counts were converted to FPKM to analyze the gene expression levels. Using Pearson’s correlation coefficient, we analyzed the relationship among the samples.

### 4.14. Gene Differential Expression Analysis

Based on the above steps, we obtained gene expression levels for a total of 12 samples from 4 coral species (Appendix A). Orthologous transcripts between each of the two corals were searched by orthofinder v2.5.4 (David M. Emms, Oxford, UK) [161]. To make the expression levels of orthologous genes comparable between different species, we used two methods (median [138] and scbn [139]) to normalize their expression levels. Both of them were based on a group of conserved orthologous genes among different species. Therefore, we selected orthologous transcripts that were uniquely present in all four coral species as conserved orthologous genes. Then, we used the median method by assessing their median expression levels in each species among the genes with expression values in the interquartile range for different species and deriving the scaling factor that adjusted those median values to a common value. Meanwhile, we used these conserved orthologous genes by minimizing the deviation between the empirical and nominal type-I errors to achieve the optimal scaling factor in the scbn method. Accordingly, the normalized expression matrix and the *p*-value of each pair of orthologous transcripts can be obtained. Orthologous transcripts with *p*-value < 10^−6^ and |log2(FoldChange)| > 2 or 10 were regarded as DEGs.

### 4.15. Phylogenetic Analysis

To construct the phylogenetic tree, 4 of our own reef-building corals and 36 other species were selected for the study, for a total of 17 clades, with the exception of the Porifera clade for which no further protein model species could be found;2 two or more representative species per clade were selected to reduce the contingency of results (Table 5 and detailed in Appendix A). The confirmed amino acid sequences of circadian clock genes of *Homo sapiens*, *Mus musculus*, *Danio rerio*, and *Drosophila melanogaster* were used as query sequences to search for orthologs in the protein models of the above species using the BLASTp algorithm at the local BLAST+ 2.13.0 (parameters: e value = 1 × 10^−5^). The sequences alignment was performed using MAFFT v7.505 (Kazutaka Katoh, Tokyo, Japan) with default parameters [162]. The gaps were removed to obtain conserved domains by trimAl v1.4.rev15 (Salvador Capella-Gutiérrez, Barcelona, Spain) with default parameters [163]. The phylogenetic trees were constructed using the maximum likelihood method by IQ-TREE 2.0.7 (Bui Quang Minh, Canberra, ACT, Australia) with the parameters: −m MFP −B 1000 −alrt 1000 (“−m MFP” means that it can automatically determine the best-fit model for data, “−B 1000” specifies 1000 replicates for the ultrafast bootstrap, and “−alrt 1000” specifies the number of bootstrap replicates for SH-aLRT where 1000 is the minimum number recommended) [164,165,166]. iTOL v6.3.2 (Ivica Letunic, Heidelberg, Germany) [167] was used to visualize the tree and Jalview 2.11.2.4 (Andrew M. Waterhouse, Cambridge, MA, USA) was used to show the conserved domains [168]. The names (abbreviations) of the 40 species were as follows: *H. sapiens* (*Hsa*), *M. musculus* (*Mmu*), *D. rerio* (*Dre*), *Gadus morhua* (*Gmo*), *Ciona intestinalis* (*Cin*), *Styela clava* (*Scl*), *B. floridae* (*Bfl*), *B. belcheri* (*Bbe*), *Strongylocentrotus purpuratus* (*Spu*), *Lytechinus variegatus* (*Lva*), *Saccoglossus kowalevskii* (*Sko*), *Ptychodera flava* (*Pfl*), *Caenorhabditis elegans* (*Cel*), *Brugia malayi* (*Bma*), *D. melanogaster* (*Dme*), *Cotesia glomerata* (*Cgl*), *C. gigas* (*Cgi*), *Pecten maximus* (*Pma*), *Schistosoma mansoni* (*Sma*), *Paragonimus kellicotti* (*Pke*), *P. damicornis* (*Pda*), *P. verrucosa* (*Pve*), *A. muricata* (*Amu*), *M. foliosa* (*Mfo*), *A. digitifera* (*Adi*), *S. pistillata* (*Spi*), *Orbicella faveolata* (*Ofa*), *N. vectensis* (*Nve*), *E. diaphana* (*Edi*), *Hydra vulgaris* (*Hvu*), *Hydra viridissima* (*Hvi*), *Morbakka virulenta* (*Mvi*), *Alatina alata* (*Aal*), *Aurelia aurita* (*Aau*), *Cassiopea xamachana* (*Cxa*), *Amphimedon queenslandica* (*Aqu*), *Mnemiopsis leidyi* (*Mle*), *Pleurobrachia bachei* (*Pba*), *Paramecium tetraurelia* (*Pte*), and *Ichthyophthirius multifiliis* (*Imu*).

## 5. Conclusions

The sequencing and analysis of the full-length transcriptome maps of dominant reef-building coral holobionts contributed to a more in-depth understanding of coral physiology. Related insights improve our understanding of the evolution of circadian rhythms and of holobiosis, and they provide a foundation for further work to protect or even manipulate coral skeleton production or symbiosis to promote the survival of these important organisms. These results provide support for remodeling reef-building corals through advanced gene toolkits under current global climate change and ecological deterioration.

## Figures and Tables

**Figure 1 ijms-23-11135-f001:**
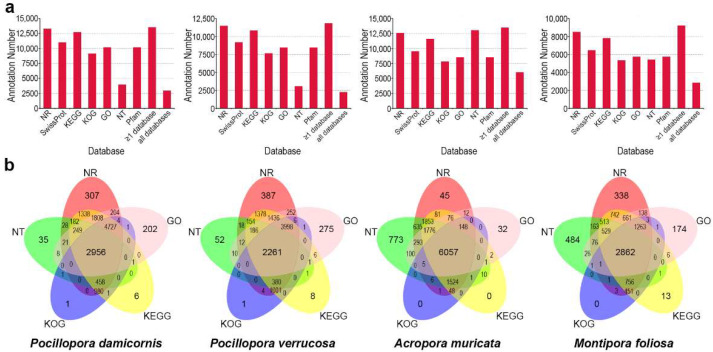
Summary of coral gene-function annotation. (**a**) Statistics of the annotation results of four reef-building corals, excluding Symbiodiniaceae sequences, in the NR, Swiss-Prot, KEGG, KOG, GO NT, and Pfam databases. The horizontal axis represents the different functional databases, and the vertical axis represents the number of sequences annotated in different functional databases, at least one database, and all databases. (**b**) Venn plots of the number of annotated sequences of four reef-building corals excluding Symbiodiniaceae sequences obtained using the NR, KEGG, KOG, GO, and NT databases. The sum of the numbers in each large circle represents the number of transcripts annotated in one database, and the overlapping circles indicate the numbers of transcripts annotated in these databases simultaneously.

**Figure 2 ijms-23-11135-f002:**
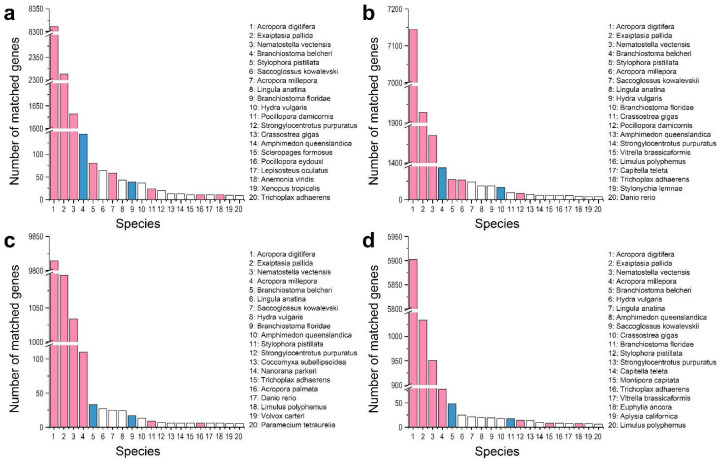
NR database annotations of corals. The top-20 species with the greatest number of top sequence hits for *P. damicornis* (**a**), *P. verrucosa* (**b**), *A. muricata* (**c**), and *M. foliosa* (**d**) are shown. The horizontal axis represents the species ID, and the vertical axis represents the number of coral unigenes annotated for different species. The pink columns represent species belonging to Anthozoa. The blue columns represent species belonging to the genus *Branchiostoma*.

**Figure 3 ijms-23-11135-f003:**
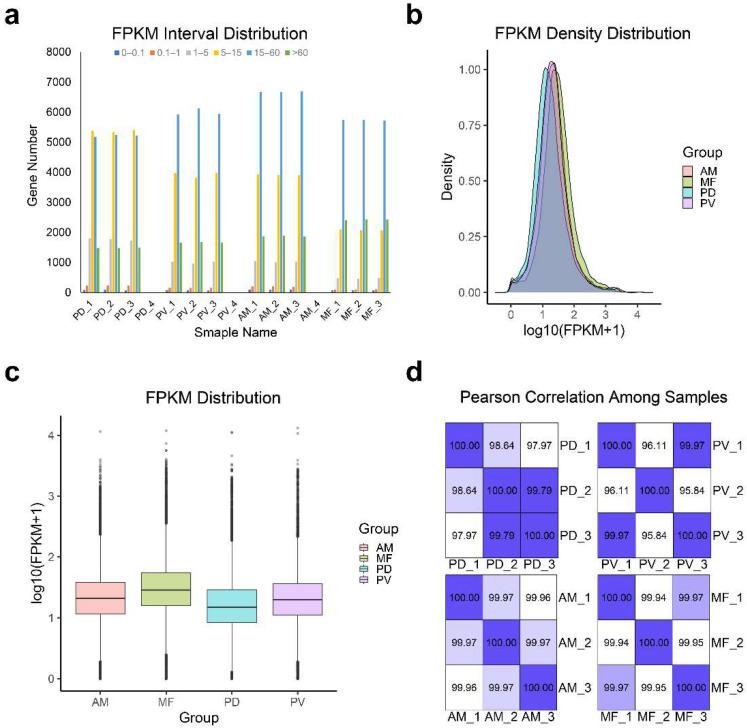
Summary of the coral gene expression level analysis. (**a**) FPKM interval distribution. The horizontal axis shows the sample name, different colors represent FPKM intervals, and the vertical axis represents the number of genes in each interval. (**b**) FPKM density distribution. The horizontal axis represents the log10 (FPKM + 1) values, and the vertical axis represents the density of genes with different expression levels. (**c**) FPKM box plot. The horizontal axis shows the sample name, and the vertical axis represents the log10 (FPKM + 1) values. Each box plot shows five statistical values, including the maximum, upper quartile, median, lower quartile, and minimum, from top to bottom. (**d**) Pearson’s correlation among samples. The closer the value is to 1, the better the correlation. Pd: *P. damicornis*; Pv: *P. verrucosa*; Am: *A. muricata*; and Mf: *M. foliosa*.

**Figure 4 ijms-23-11135-f004:**
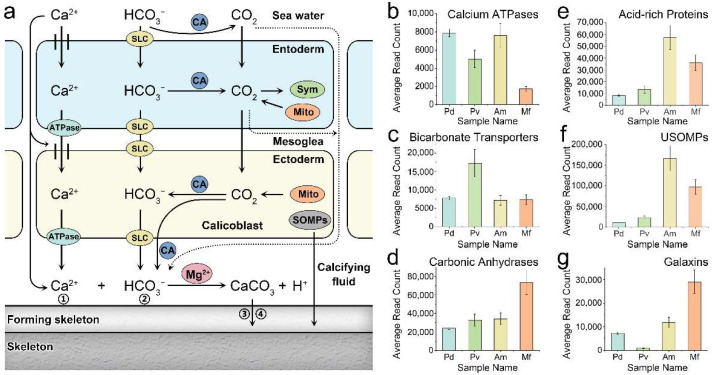
Mechanism of coral biomineralization and related gene expression levels. (**a**) The mechanism of reef-building coral biomineralization. (1) Ca^2+^ transport by calcium ATPase or diffusion. (2) CO_2_ can be converted into HCO_3_^−^ by CA and then exits the cells via bicarbonate transporters (i.e., SLC4 and SLC26). (3) Coral acid-rich proteins and Mg^2+^ can promote crystal nucleation, determine growth axes, and control crystal growth. (4) The bioprecipitation of aragonite crystals in corals requires additional skeletal organic matrix proteins (USOMPs, galaxins, and alpha IV collagen). The average expression levels of genes belonging to calcium ATPases (**b**), bicarbonate transporters (**c**), carbonic anhydrases (**d**), acid-rich proteins (**e**), USOMPs (**f**), and galaxins (**g**). The horizontal axis shows the sample name, and the vertical axis represents the average read-count values of each set of genes. ATPase: calcium ATPase; SLC: solute carriers 4 (SLC4) and 26 (SLC26); CA: carbonic anhydrase; Sym: Symbiodiniaceae; Mito: mitochondrion; and SOMPs: coral acid-rich proteins, alpha IV collagen, galaxins, and uncharacterized skeletal organic matrix proteins (USOMPs). The solid lines represent definite paths and the dashed lines represent possible paths.

**Figure 5 ijms-23-11135-f005:**
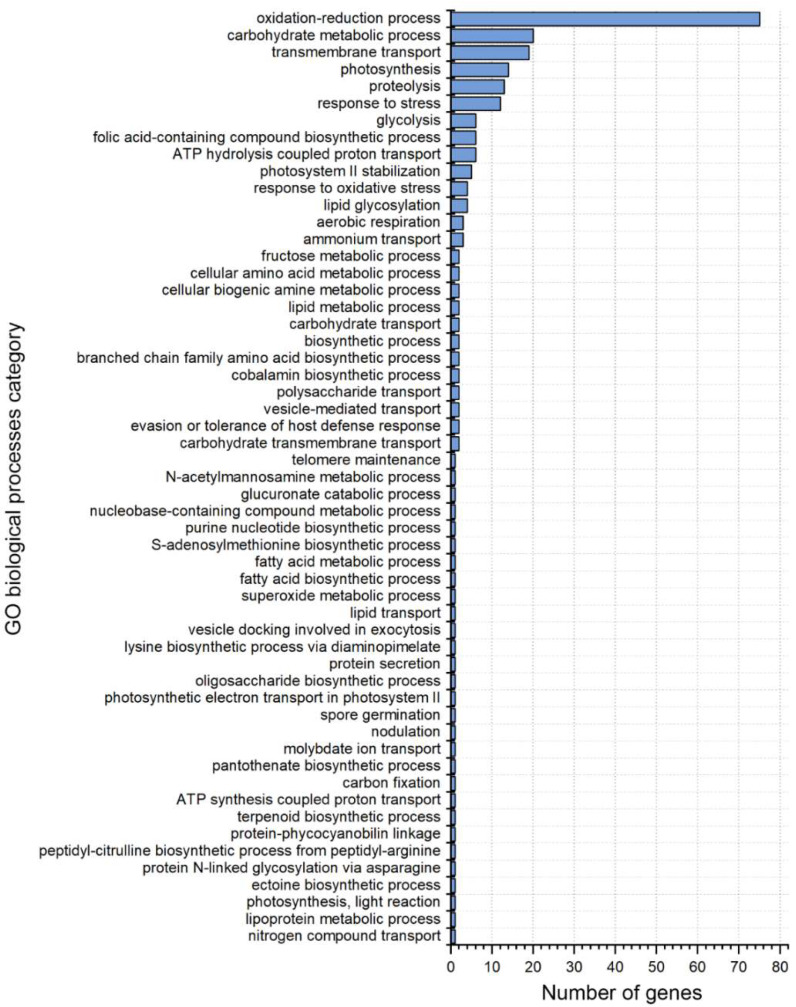
Bar graph of co-expressed Symbiodiniaceae sequences based on GO biological processes categories. The vertical axis represents the GO terms, and the horizontal axis represents the number of transcripts annotated to the terms (including sub-terms of the terms).

**Figure 6 ijms-23-11135-f006:**
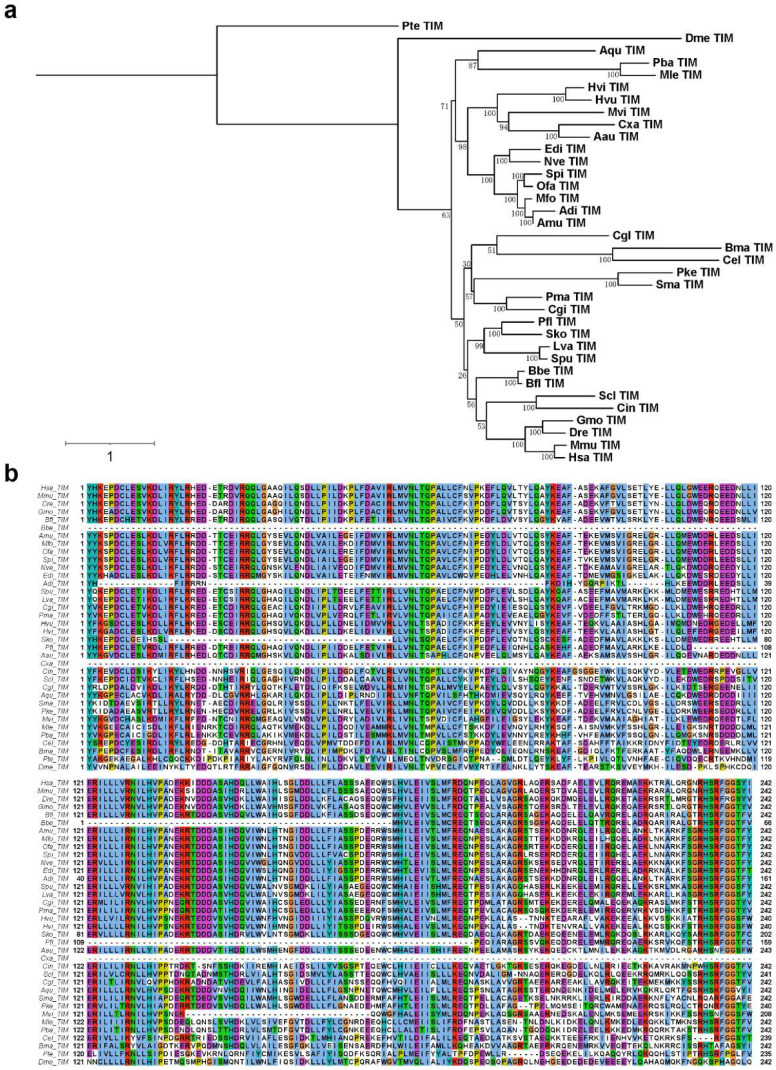
*Tim* gene phylogenetic analysis. (**a**) Phylogenetic tree of *tim* genes based on the maximum likelihood method with best-fit model selection. Branch lengths are optimized by maximum likelihood on original alignment and the numbers are bootstrap supports (%). (**b**) Partial conserved domain (pfam04821) of *tim* genes among different species. The complete conserved domain is shown in Appendix A.

**Table 1 ijms-23-11135-t001:** Sequence data of four reef-building corals in the NCBI database.

NO.	BioProject	Data Volume, Gb	Data Volume, Mbytes	Library Strategy	Platform	Model
*P. damicornis*						
1	PRJNA227785	27	17,496	RNA-Seq	Illumina	Genome Analyzer IIx
2	PRJNA299443	24	10,069	RNA-Seq	Illumina	HiSeq 4000
3	PRJNA327142	10	3750	RNA-Seq	Illumina	HiSeq 4000
4	PRJNA433950	60	23,545	RNA-Seq	Illumina	HiSeq X Ten
5	PRJNA435620	67	26,334	RNA-Seq	Illumina	HiSeq X Ten
6	PRJNA306839	18	8574	RNA-Seq	Illumina	MiSeq
7	PRJEB26650	45	21,254	RNA-Seq	Illumina	HiSeq 2500
**8**	**PRJNA544778**	**1348**	**0.41T**	**RNA-Seq**	**PacBio** **Illumina**	**Sequel** **II** **HiSeq X Ten**
9	PRJNA545379	67	27,278	RNA-Seq	Illumina	HiSeq 2000
10	PRJNA611041	258	90,416	RNA-Seq	Illumina	HiSeq 1500
*P. verrucosa*						
1	PRJNA552592	15	7782	RNA-Seq	Illumina	HiSeq 2500
2	PRJNA551401	108	47,540	RNA-Seq WGS	Illumina	HiSeq4000 HiSeq2500
**3**	**PRJNA544778**	**1348**	**0.41T**	**RNA-Seq**	**PacBio** **Illumina**	**Sequel** **II** **HiSeq X Ten**
*A. muricata*						
**1**	**PRJNA544778**	**1348**	**0.41T**	**RNA-Seq**	**PacBio** **Illumina**	**Sequel** **II** **HiSeq X Ten**
2	PRJDB8519	1785	1.09T	WGS	Illumina	HiSeq 2500
3	PRJDB5633	110	63,481	WGS	Illumina	HiSeq 2500
*M. foliosa*						
**1**	**PRJNA544778**	**1348**	**0.41T**	**RNA-Seq**	**PacBio** **Illumina**	**Sequel** **II** **HiSeq X Ten**

Only datasets of ≥10 Gb coral sequences (i.e., excluding Symbiodiniaceae sequences) are included. The bold lines indicate data generated in this paper.

**Table 2 ijms-23-11135-t002:** Statistics of PacBio Iso-seq data.

Sample Name	*P. damicornis*	*P. verrucosa*	*A. muricata*	*M. foliosa*
Polymerase reads (Gb)	25.34	27.8	22.32	21.44
Subreads (Gb)	24.43	26.52	21.36	20.24
CCS ^1^ (Number)	334,111	292,565	341,490	305,153
FLNC ^2^ (Number)	245,504	249,577	273,822	238,498
Polished consensus (Number)	20,994	24,860	31,571	26,455
Transcripts (Number)	20,609	24,174	31,242	25,460
Unigenes (Number)	14,167	12,822	13,800	10,905
Mean length (bp)	2668	2092	1918	1472
Minimum length (bp)	51	107	183	69
Maximum length (bp)	9888	7861	6991	5942
N50 ^3^ (bp)	2954	2313	2126	1678
N90 ^3^ (bp)	1793	1346	1230	905

^1^ CCS: circular consensus sequence. ^2^ FLNC: full-length non-chimera sequences. ^3^ N50 or 90: the length for which the collection of all subreads of that length or longer contains at least 50% or 90% of the total of the lengths of the subreads.

**Table 3 ijms-23-11135-t003:** The top-five species with the genes most similar to the four investigated corals.

Rank	Species	Gene Number ^1^	Percentage ^2^
*P. damicornis*			
1	*Acropora digitifera*	8310	62.66%
2	*Exaiptasia pallida*	2313	17.44%
3	*Nematostella vectensis*	1632	12.31%
4	*Branchiostoma belcheri*	145	1.09%
5	*Stylophora pistillata*	80	0.60%
*P. verrucosa*			
1	*Acropora digitifera*	7144	62.27%
2	*Exaiptasia pallida*	1928	16.80%
3	*Nematostella vectensis*	1475	12.86%
4	*Branchiostoma belcheri*	86	0.75%
5	*Stylophora pistillata*	54	0.47%
*A. muricata*			
1	*Acropora digitifera*	9814	78.14%
2	*Exaiptasia pallida*	1098	8.74%
3	*Nematostella vectensis*	1034	8.23%
4	*Acropora millepora*	110	0.88%
5	*Branchiostoma belcheri*	33	0.26%
*M. foliosa*			
1	*Acropora digitifera*	5902	69.44%
2	*Exaiptasia pallida*	1034	12.16%
3	*Nematostella vectensis*	951	11.19%
4	*Acropora millepora*	78	0.92%
5	*Branchiostoma belcheri*	48	0.56%

^1^ The number of coral genes that are annotated with the species listed in column 2 by NR database. ^2^ The gene number listed in column 3 as % of all in coral.

**Table 4 ijms-23-11135-t004:** Average read-count values of key biomineralization-related genes in four investigated corals.

Class		Protein Name	*P. damicornis*	*P. verrucosa*	*A. muricata*	*M. foliosa*
Calcium ATPase		Plasma membrane calcium-transporting ATPase 1	0.00	0.00	5081.58	0.00
		Plasma membrane calcium-transporting ATPase 2	1716.33	0.00	0.00	1171.67
		Plasma membrane calcium-transporting ATPase 3	0.00	835.00	1049.33	528.33
		Plasma membrane calcium-transporting ATPase 4	1596.00	1549.33	0.00	0.00
		Plasma membrane calcium ATPase	4570.67	2626.67	0.00	0.00
		Calcium-transporting ATPase type 2C member 1	0.00	0.00	1466.92	0.00
Bicarbonate Transporter		Solute carrier 4 (SLC4)	5580.66	8157.00	6321.27	5829.46
		Solute carrier 26 (SLC26)	2160.45	9072.61	850.67	1447.00
Alpha Carbonic Anhydrase		Carbonic anhydrase 1	757.33	71.33	0.00	0.00
		Carbonic anhydrase 2	22,775.35	32,853.41	31,611.00	66,451.10
		Carbonic anhydrase 3	508.33	0.00	0.00	0.00
		Carbonic anhydrase 12	0.00	0.00	2580.00	6691.27
Acidic Proteins		Skeletal aspartic acid-rich protein 1 (SAARP1)	6423.17	11,894.42	12,233.75	3600.29
		Skeletal aspartic acid-rich protein 2 (SAARP2)	0.00	341.41	1424.33	0.00
		Acidic skeletal organic matrix protein (Acidic SOMP)	1831.00	1191.00	6063.00	4909.22
		Secreted acidic protein 1 (SAP1)	0.00	0.00	0.00	5512.38
		Secreted acidic protein 2 (SAP2)	0.00	0.00	14,584.00	8046.54
		Aspartic and glutamic acid-rich protein	0.00	0.00	23,130.97	13,798.33
Unique Uncharacterized Proteins	Uncharacterized skeletal organic matrix protein	Uncharacterized skeletal organic matrix protein-1 (USOMP-1)	0.00	0.00	598.72	0.00
		Uncharacterized skeletal organic matrix protein-2 (USOMP-2)	232.00	133.33	1202.33	4982.33
		Uncharacterized skeletal organic matrix protein-3 (USOMP-3)	2138.94	0.00	1768.33	4098.34
		Uncharacterized skeletal organic matrix protein-4 (USOMP-4)	0.00	0.00	21,585.00	0.00
		Uncharacterized skeletal organic matrix protein-5 (USOMP-5)	7592.00	18,666.81	7931.45	2664.33
		Uncharacterized skeletal organic matrix protein-6 (USOMP-6)	0.00	0.00	131,300.52	80,894.89
		Uncharacterized skeletal organic matrix protein-7 (USOMP-7)	819.49	206.00	2158.33	4538.67
		Uncharacterized skeletal organic matrix protein-8 (USOMP-8)	0.00	4156.33	0.00	0.00
	Galaxin	Galaxin	7224.57	862.92	6324.67	19,549.24
		Galaxin2	0.00	0.00	5482.00	9479.72
Collagen alpha-6(VI)		Collagen alpha-6 (VI) chain	0.00	284.67	3649.33	1082.93

**Table 5 ijms-23-11135-t005:** Presence of key circadian clock gene homologues in representative species.

Lineage	Key Circadian Clock Genes										
	*cry1*	*cry2*	*Clock*	*Npas2*	*cyc*	*Arntl* (*Bmal1*)	*Arntl2 (Bmal2*)	*per1*	*per2*	*per3*	*tim*
Mammalia	Y	Y	Y	Y	-	Y	Y	Y	Y	Y	Y
Actinopterygii	Y	Y	Y	Y	-	Y	Y	Y	Y	Y	Y
Ascidiacea	-	-	-	-	-	Y	-	-	-	-	Y
Cephalochordata	Y	-	-	Y	-	Y	-	Y	-	-	Y
Echinoidea	Y	Y	Y	-	-	Y	-	-	-	-	Y
Enteropneusta	-	-	Y	-	-	Y	-	-	-	-	Y
Nematoda	-	-	-	-	-	Y	-	-	-	-	Y
Insecta	Y	-	Y	-	Y	-	-	Y	-	-	Y
Bivalvia	Y	-	Y	-	Y	-	-	Y	-	-	Y
Platyhelminthes	-	-	-	-	-	Y	-	-	-	-	Y
Anthozoa	Y	Y	Y	Y	Y	Y	-	-	-	-	Y
Hydrozoa	Y	-	-	-	-	Y	-	-	-	-	Y
Cubozoa	-	-	-	Y	-	Y	-	-	-	-	Y
Scyphozoa	Y	-	-	-	-	Y	-	-	-	-	Y
Porifera	-	Y	Y	-	Y	-	-	-	-	-	Y
Ctenophora	Y	-	Y	-	-	Y	-	-	-	-	Y
Ciliophora	-	-	-	-	-	-	-	-	-	-	Y

“Y” means yes, which means the species contains this gene homologue, while “-“ means the opposite.

## Data Availability

The data (reef-building coral holobionts full-length and short-read transcriptome sequencing raw data) presented in this study are openly available in SRA at https://www.ncbi.nlm.nih.gov/sra (accessed on 18 September 2022). The reference numbers are SRR9613489 and SRR12904788-90 for *Pocillopora damicornis*; SRR12963486, SRR12959228, and SRR12959239-40 for *Pocillopora verrucosa*; SRR12963485, SRR12959195, SRR12959206, and SRR12959217 for *Acropora muricata*; and SRR12963484 and SRR12959182-4 for *Montipora foliosa*. The data (Unigene, CDS, PEP, and UTR sequences) analyzed in this study are openly available in Figshare at https://doi.org/10.6084/m9.figshare.19403021 (accessed on 18 September 2022).

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
