# Peer review of "Full-Length Transcriptome Maps of Reef-Building Coral Illuminate the Molecular Basis of Calcification, Symbiosis, and Circadian Genes"

_ijms, 2022, doi:10.3390/ijms231911135_

Round 1

Reviewer 1 Report

The manuscript of Han et al. provides a very useful transcriptomic resource on four species of reef-building corals. The authors performed PacBio sequencing to generate full-length gene models and the used Illumina sequencing to perform some expression analyses. While the sequence data presented by Han et al. is unquestionably valuable and should be published, some of the analyses they performed are problematic, and will have to be re-run in order for the paper to be publishable. Also, the biological interpretations of some of the data need to be corrected. In several cases, the authors start with suggesting a highly dubious explanation of their observations in the Results, and then, in the Discussion, they talk about it as if it was a proven fact, using the words “we confirmed” and suchlike. Below are my specific comments:

Major:

1. The authors use the number of annotated similar sequences as a measure for the evolutionary distance. That's a very creative method, but not a very robust one. Otherwise, one would be forced to conclude that P. damicornis and P. verrucosa are phylogenetically closer to Branchiostoma than to Stylophora, which is, obviously, nonsense. In general, while performing this “sanity check” to see that your transcripts really come from a cnidarian rather than from some contaminant is useful, any evolutionary conclusions based on this are impossible. If Han et al. really want to re-evaluate the phylogenetic position of their four corals using full transcriptome data, they need to select orthologous protein sequences from each of the species they analyse, concatenate them and construct phylogenetic trees. However, since the phylogenetic position of the four sequenced corals does not seem to be disputed, I suggest skipping this analysis (including Fig. 2) entirely due to a complete lack of any useful import. Obviously, coral sequences hit sequences from other cnidarians as well as slowly evolving Bilateria.

2. The whole coral-amphioxus convergence part is highly flawed. Han et al. write that continued occurrence of Branchiostoma hits suggests that “the presence of intracellular algal cells in both reef-building corals and amphioxus probably impacts the component pattern of the gene expression profiles of the host, leading to a convergence of adaptive evolution”. There is absolutely no way this assumption can be made based on the evidence the authors show. It is way more likely that the reoccurrence of amphioxus hits reflects the fact that coral and amphioxus protein sequences are slow-evolving. So, among Bilateria, the highest similarity to amphioxus is expected. In the Discussion, the authors also make a big point of the convergence between corals and amphioxus due to the presence of algal endosymbionts.  They go as far as stating that “Taken together, our findings confirm that there are some evolutionary convergences between coral and amphioxus that have been triggered by interactions with intracellular algal cells.” I do not doubt that some convergence at the level of gene expression may be expected, since intracellular symbiont uptake poses the same challenges in different models, but I do not think that Han et al. show any data supporting this. The only convincing way to demonstrate that amphioxus and coral cells with endosymbionts have convergent features is: i) single-cell RNA-Sequencing and comparing the transcriptomes of the particular cell types carrying endosymbionts in corals and amphioxus, ii) showing that same genetic sets are upregulated in these cells, and, finally, iii) showing that these genes are functionally involved in facilitating symbiosis (e.g. knocking them out results in the symbiosis loss). Any argument like “the four reef-building corals investigated herein and published coral gene expression profiles all include gene repertoires for innate immunity” is not valid. Non-endosymbiotic animals also have innate immunity. I suggest removing the whole endosymbiosis section from the Results and from the Discussion. If the authors are really determined to write something about symbiosis using available bulk data anyway, they should compare the expression of orthologous genes between their symbiotic corals, bleached corals of the same species, and non-symbiotic anthozoans like Nematostella (although it is a very distant relative of reef-building corals).

3. I fail to see how gene expression profile comparison made at a transcriptome-wide level is in any way informative for understanding "the ecological and evolutionary relationships". Especially since the taxonomic position and ecology of these species is quite known. According to the introduction, the ecology of the three sequenced species is nearly identical, while Pocillopora verrucosa lives in a slightly different habitat. In spite of this, two Pocillopora transcriptomes are most similar - which makes sense, because they are same genus. The problem is that these were not pairwise comparisons of the expression levels of orthologous genes performed on a genome-wide scale - such an analysis would make some sense - but the analyses presented here just produce some uninterpretable numbers characterizing expression levels, which can be plotted on a Figure, but are not very meaningful. I don't see what "evolutionary aspects" are being explained by these analyses, and why lower overall expression M. foliosa is called "adaptive". Any feature under selection in any organism is "adaptive" in some way, but as long as the authors cannot identify the reason for the difference, the statement of "adaptiveness" of something is meaningless. Expression levels documented in the other three species are no less adaptive. I suggest removing this analysis entirely.

4. Neither the Results nor the Methods section explain properly what was compared to what in the DGE analysis. Was it differential expression between the three biological replicates within each particular species? This would be a good analysis to run in order to see that the replicates are really similar (although it is anyway performed in the default output of DESeq2). However, I don't think this was what the authors did here. Based on their Venn diagrams, I think they looked at differential expression between the species, however, genes were compared based on their "annotation result in the NR database". This cannot be done this way. You can only compare orthologous sequences from different species. You can use e.g. OMA (Altenhoff, A. M. et al. OMA orthology in 2021: website overhaul, conserved isoforms, ancestral gene order and more. Nucleic Acids Research 49, D373–D379 (2021).), OrthoFinder (Emms, D.M., Kelly, S. OrthoFinder: phylogenetic orthology inference for comparative genomics. Genome Biol 20, 238 (2019).) or just simple reciprocal BLASTP (not against NR! but between the 4 corals you sequenced). Then you have to discard everything, which is not orthologs, and run DESeq2 on these orthologs only. All the downstream analyses as well as the discussion of the DGE results needs to be re-written. Also, an explanation of how the normalization was performed to make the libraries from different species comparable needs to be included.

5. Han et al. make conclusions about the biochemistry and physiology of the four corals they sequenced based purely on transcriptomic data. For example, they write “Among the four investigated corals, M. foliosa has the lowest apparent capacity to transport calcium ions to calcifying fluid because of the low expression of calcium ATPases”, and there is a big section in the Discussion about this. I do not think that such conclusions can be made. At the protein level, we know nothing about the activity and stability of the Ca(2+) ATPases in M. foliosa in relation to the same proteins in the other three coral species, which is what matters in the end. The assumptions the authors make are not necessarily wrong, however, their data do not allow to make them. Such statements require biochemical and physiological analyses, not transcriptomics. Han et al. conclude their calcification section with a statement, which has no connection to their results but can only be based on paleontological evidence: “Combined with the observation of growth patterns, it can be inferred that after the Tertiary Period, the Complexa clade corals developed to occupy more space (as observed A. muricate and M. foliosa), whereas the Robusta clade corals tended to form more dense colonies (as observed in P. verrucosa).”  

6. In order to understand the distribution of the circadian genes across the animal tree, the authors appear to have run a blast search against NR database searching in a single species from a major clade and make a massive evolutionary claim about the time of the evolution of the circadian system (apparently, after the ctenophore split, although in the beginning of the introduction the authors stated that sponges were the oldest animal lineage) and about its loss in jellyfish and Turbellaria. I found these statements difficult to believe and performed a quick BLASTP search against protein models in the ctenophore Mnemiopsis (https://research.nhgri.nih.gov/mnemiopsis/sequenceserver/), scyphozoan jellyfish Aurelia (https://marinegenomics.oist.jp/aurelia_aurita/blast/), and a turbellarian Schmidtea (https://parasite.wormbase.org/Schmidtea_mediterranea_prjna12585/Tools/Blast) using Branchiostoma sequences of CRY1, CLOCK, NPAS, ARNTL1, PERIOD, and TIMELESS, as well as Nematostella CLOCK. I found potential representatives of the circadian system in all these species. Mnemiopsis had good protein models for CRY1, NPAS, CYCLE/ARNTL1, and TIMELESS. At the first glance, there were no good models for CLOCK and PERIOD. Aurelia had good hits against all these circadian proteins except PERIOD! Schmidtea had good hits for CYCLE, ARNTL1, and TIMELESS, but not for the other circadian proteins. I did not look at other clades, in which the loss of the circadian system has been postulated by the authors. However, the clear lack of rigour in the analysis makes it obvious that all these searches must be redone from scratch and all the evolutionary conclusions rigorously re-evaluated for the revised version of the manuscript. Given the abundance of sequencing information currently available, the authors should repeat their searches using more than just one representative of each clade, and using not the NR database but the genomes or transcriptomes of the species in question. After all these analyses are re-done, this part of the Results and Discussion must be re-written accordingly. I must admit that a mistake of such magnitude also undermines the credibility of the rest of the statements of the authors, and makes me think that I might have overlooked some other major mistakes in the other parts of the paper.

7. The authors perform the phylogenetic analysis of Tim. What about the other circadian proteins? Also, the only good thing about the NJ algorithm Han et al. used is that it is quick. The authors should perform a best-fit model selection followed by proper maximum likelihood analysis for all their circadian proteins. It would be useful if the authors could upload supplementary files with all the protein sequences they use for the alignments (including their names and accession numbers). Showing the multiple alignment in the main text does not appear necessary. A gene presence/absence table in the main text, which is based on a good ML tree for each of the circadian proteins (trees should go into the supplement), is perfectly sufficient.

Minor:

1. The first two sentences of the abstract are weird – it is not clear, how they are connected, and why there is a “however” between them. I suggest replacing the whole first sentence and "However, relevant" at the beginning of the second sentence by one word "Coral", so that your abstract starts with "Coral transcriptomic data largely relies on....".

2. The authors use “primitive” in several places in the text, for example in “The scleractinian corals that mainly produce coral reefs constitute one of the most primitive metazoan branches”. While cnidarian-bilaterian split is very old, the scleractinian corals aren't an old group of cnidarians. According to paleontological evidence, Scleractinia appeared in the Triassic period (https://www.sciencedirect.com/science/article/abs/pii/S0012825202001046), i.e. at the same time as the archosaurs - the ancestors of crocodiles, dinosaurs and birds. Also, amphioxus is called a “primitive deuterostome”. Yes, it has a simple morphology, its genome is extremely conserved, but its development from neurulation on is very derived. I suggest rephrasing. The authors can use "morphologically simple" or something other than “primitive”.

3. The authors write “Phagocytosis targeting primitive algae was a prerequisite for the development of food webs involving multicellular animals, as well as the origin of mitochondria and chloroplasts”. However, the origin of mitochondria or chloroplasts has nothing to do with phagocytosing algae. It is about prokaryotic organisms engulfing other prokaryotic organisms.  "Blue-green algae", which probably had to be engulfed to make chloroplasts, are not algae, but bacteria. Mitochondria most probably emerged when an Asgard-type Archea-like cell engulfed and alphaproteobacterium. The sentence in the manuscript is misleading and must be rephrased.

4. The authors write that “intracellular digestion involves direct preying by cell cilia [57]”. "Direct preying by cilia" sounds extremely strange to me. I checked reference 57, and I did not find anything about ciliary preying there. I might not be familiar enough with the phagocytosis literature, but I have never heard of an example when food - especially as large particles as algal cells - would be phagocytosed by the cilium. Can it be that you mixed them up with microvilli? Although "direct preying by a microvillus" also does not sound right.

5. The authors call sponges “the oldest extant animal lineage”. However, the phylogenetic position of sponges is not settled. There is a debate, which has been especially intense since 2008, on whether sponges or ctenophores were the earliest branching animals. Please mention this and refer to relevant papers.

6. Han et al write that “24 circadian genes in the Acropora millepora coral transcriptome have orthologues in bilaterian species”.  This is misleading because it is exactly the other way around from what the original paper says. These Acropora genes are thought to be circadian because they were found by BLAST-ing 24 bilaterian circadian genes against Acropora transcriptome, and not because these Acropora genes were shown to act in the circardian clock in Acropora. In the Levy et al., 2007 paper, cycling expression has been demonstrated for the Cry genes. There are no published functional data on potential circadian genes in Cnidaria.

7. The authors always call their transcriptomes full-length. This is not entirely correct. Han et al. used SMARTer PCR cDNA synthesis kit. It effectively enriches for full-length cDNAs, but it does not give exclusively full-length products. I suggest calling it a "full-length-enriched raw transriptomic sequencing". Also the next sentence about removing reads of less than 50 bp makes much more sense if one does not state that all the reads are full-length.

8. When discussing the benefits of the PacBio quality and discrepancy in the gene model number between short read-based transcriptomes and PacBio, the authors should mention that their gene number is bound to be an underestimation since they sequenced only one developmental stage.

9. In Table 3, the authors list “ top five species with the most similar genes to the four investigated corals”. They need to define "most similar genes". What is "percentage"? Is it nucleotide identity? Amino acid identity in the translated proteins? Amino acid similarity? (Although, as mentioned in the Major comments, I would discard this section completely!).

10. Han et al. repeatedly use the term Actinozoa in several places of the manuscript. They mixed it up with Anthozoa, and should correct it. Actinozoa is not a valid taxon. It is an entirely useless systematic term coined by de Blainville in 1834 to describe animals with radial symmetry. Actinozoa united unrelated groups of animals such as cnidarians, echinoderms, rotifers and bryozoans. Using it in 2022 does not make any sense (as well as Radiata, which is also not a valid term also used by the authors).

11. The authors write “consequently, the phylogenetic analysis suggested that in the early metazoan stage, there were four key gene members of the circadian clock gene regulatory network”. The authors should define an "early metazoan stage"? Do they mean the last common ancestor of all multicellular animals or some other node on the tree?

12. I do not see any evidence for the author’s statement that higher similarity of the different Symbiodinium transcriptomes than the coral host transcriptomes “suggested that there may be some association between marine animal intracellular symbiosis and the intracellular digestion of algal cells.” Rather it reflects the fact that the internal environment of coral cells carrying the algae in all four species is probably quite similar, which makes the Symbiodinium transcriptomes less variable than the coral transcriptomes.

13. Since the presence of the complement of circadian genes has been previously shown in anthozoans by other authors, the statement that this paper is “providing a molecular basis for the study of the evolutionary origin of the animal circadian clock system” is clearly taking credit for something, which has been shown earlier by others.

14. The Conclusions part of the manuscript does not contain any useful information and can be skipped.

15. The numbers of the SRA projects mentioned in the Data Availability cannot be found on NCBI. Unless the authors blocked them until the publication date, they should double-check with the NCBI that the links are active.

Author Response

Dear Ms. Hang,

Thank you for giving us the opportunity to submit a revised draft of the manuscript “Full-length transcriptome maps of reef-building coral illuminate the molecular basis of calcification, symbiosis, and circadian genes” for publication in International Journal of Molecular Sciences. We appreciate the time and effort that you and the reviewers dedicated to providing feedback on our manuscript and are grateful for the insightful comments on and valuable improvements to our paper. We have incorporated most of the suggestions made by the reviewers. Those changes are highlighted within the manuscript. Please see the attachment, for a point-by-point response to the reviewers’ comments and concerns. All page and line numbers refer to the revised manuscript file hiding the tracked changes.

Sincerely,

Chunpeng He

State Key Laboratory of Bioelectronics, School of Biological Science and Medical Engineering, Southeast University, Nanjing 210096, China.

e-mail: cphe@seu.edu.cn

Zuhong Lu

State Key Laboratory of Bioelectronics, School of Biological Science and Medical Engineering, Southeast University, Nanjing 210096, China.

e-mail: zhlu@seu.edu.cn

Reviewer 2 Report

This is an interesting work by Han et al looking at Full-Lenght trancriptome of 4 coral species and discuss no the biomineralization, symbiosis and circadian genes.

In my opinion the study has been executed well. 

I have few questions

1. I don't understand the part about Montipora capricornis? it is confusing. It is only mentioned toward the end. Was this Montipora originally sampled and authors thought as M. foliosa and then it also consisted of M. capricornis?

2. Why did the authors decide to keep the corals in the laboratory rather  than just using the fresh samples for the analysis? How long were the corals maintained in the tanks before they were used for RNA extraction? 

3. When you discuss about the similarity of gene expression in Symbiodiniaceae between coral hosts, isn't it necessary to discuss about what Symbiodiniacee species these corals host in Xisha? I think this is an important point to consider

4. How does the percentage of sequences for Symbiodniaceae obtained in this work relate to the previous studies? Becasue this is one of the points you make in the introduction regarding short reads.

5. Why you focussed on biomineralization, circadian gene and symbiosis?

Minor comments

There are spelling mistakes with respect to species - please pay attention - for example A. muricate

Author Response

(The authors gave the same response as above.)

Round 2

Reviewer 1 Report

The authors have addressed most of the points, which made the manuscript much better. However, there is still one serious problem preventing publication:

Differential expression analysis has not been improved in the revision. If I understood correctly what the authors did – it is not clearly described anywhere in the manuscript – the whole differential expression part is faulty. The method Han et al. seem to have chosen for the differential expression analysis between species just cannot be used. From their response letter, it looks like Han et al. combined their 4 full-length transcriptomes into a single reference transcriptome, mapped the Illumina reads from the four corals against this fused reference, and analysed differential expression. This method is bound to produce artefacts and give a much higher number of differentials than there is in reality because reads, which belong to orthologous genes from the four species, would map to different unigenes in the reference transcriptomes. For example, let us assume that TIM is present in all four sequenced coral species, and after normalization we have 100 TIM reads from each species. Once we map these reads to the reference transcriptome, which is in fact a mixture of four transcriptomes, TIM from P. verrucosa will, simply due to higher sequence similarity, map to one unigene in the combined reference, TIM from P. damicornis – to another, TIM from M. foliosa – to the third, and TIM from A. muricata – to the fourth. Thus, the mapping of the short reads belonging to ortholgous genes from four coral species to different unigenes in the reference will result in DESeq2 falsely recognizing these genes as differentially expressed, although they are not! This, rather than anything biologically relevant, must be the reason why Han et al. get such incredible numbers of differentially expressed genes with log2(FoldChange)>10 for Acropora: they found 10263 DEGs out of 10905 unigenes (25460 transcripts) they describe for this species! This paper cannot be published unless the authors rectify this as I specified in the 4th point of my review for the original version of the manuscript, namely they must first identify the orthologs, then compare whether these orthologs are differentially expressed in the four species.

Minor points

1. The newly added text (marked in red in the pdf) must be proof-read: the English is not great.

2. On Fig. 3, the authors have “Smaple Name” instead of “Sample name”.

Author Response

(The authors gave the same response as above.)

Round 3

Reviewer 1 Report

I am surprised the authors did not provide even a rough overview of what were these orthologous genes, which they found to be differentially expressed in different corals. However, the paper appears to be technically sound and can be published.